# Gdf3 is required for robust Nodal signaling during germ layer formation and left-right patterning

Jose L Pelliccia[1], Granton A Jindal[1,2,3], Rebecca D Burdine[1]*

[1]Department of Molecular Biology, Princeton University, Princeton, United States; [2]Department of Chemical and Biological Engineering, Princeton University, Princeton, United States; [3]The Lewis-Sigler Institute for Integrative Genomics, Princeton University, Princeton, United States

**Abstract** Vertebrate embryonic patterning depends on signaling from Nodal, a TGFβ superfamily member. There are three Nodal orthologs in zebrafish; *southpaw* directs left-right asymmetries, while *squint* and *cyclops* function earlier to pattern mesendoderm. TGFβ member Vg1 is implicated in mesoderm formation but the role of the zebrafish ortholog, Growth differentiation factor 3 (Gdf3), has not been fully explored. We show that zygotic expression of *gdf3* is dispensable for embryonic development, while maternally deposited *gdf3* is required for mesendoderm formation and dorsal-ventral patterning. We further show that Gdf3 can affect left-right patterning at multiple stages, including proper development of regional cell morphology in Kupffer's vesicle and the establishment of *southpaw* expression in the lateral plate mesoderm. Collectively, our data indicate that *gdf3* is critical for robust Nodal signaling at multiple stages in zebrafish embryonic development.
DOI: https://doi.org/10.7554/eLife.28635.001

*For correspondence:
rburdine@princeton.edu

**Competing interests:** The authors declare that no competing interests exist.

## Introduction

Transforming Growth Factor-β (TGFβ) superfamily proteins Nodal and Bone Morphogenetic Protein (Bmp) are secreted ligands that play essential roles in development, including maintaining pluripotency of embryonic stem cells and in the patterning of the early embryo (*Morikawa et al., 2016*). The well-conserved Nodal ligand binds to a heteromeric complex of receptors that phosphorylate Smad2/3, which enables its association with Smad4 (*Hata and Chen, 2016*). This Smad complex then interacts with transcription factors in the nucleus to activate the expression of target genes. In embryogenesis, Nodal signaling is required for the formation of the mesoderm and endoderm germ-layers (*Iannaccone et al., 1992*; *Conlon et al., 1994*; *Jones et al., 1995*; *Feldman et al., 1998*) and for proper establishment of left-right (L-R) asymmetry (*Levin et al., 1995*; *Collignon et al., 1996*; *Lowe et al., 1996*; *Sampath et al., 1997*; *Long et al., 2003*; *Grimes and Burdine, 2017*). In zebrafish, the Nodal orthologs Nodal-related1 (Ndr1; previously known as Squint) and Nodal-related 2 (Ndr2; previously known as Cyclops) primarily function in mesoderm and endoderm formation (*Feldman et al., 1998*) while a third ortholog, Southpaw (Spaw), is essential for proper L-R patterning (*Long et al., 2003*). The Bmp signaling pathway shares some common components with the Nodal pathway except Bmp binds to a different heteromeric complex of receptors that phosphorylates Smad1/5/8, enabling their association with Smad4 and ultimately activating Bmp-specific target genes (*Hata and Chen, 2016*). In the embryo, a ventral-to-dorsal decreasing Bmp activity gradient functions to specify ventral fates (*De Robertis, 2009*; *Langdon and Mullins, 2011*). Overall, the correct temporal and spatial signaling combinations from the Nodal (*van Boxtel et al., 2015*; *Dubrulle et al., 2015*; *Sako et al., 2016*) and Bmp (*De Robertis, 2009*; *Schmid et al.,*

*2000*) pathways are essential along the dorsal-ventral axis for organization of the vertebrate embryo (*Xu et al., 2014*; *Thisse and Thisse, 2015*).

An additional TGFβ ligand, Vg1, has also been implicated in both mesoderm formation (*Thomsen and Melton, 1993*; *Kessler and Melton, 1995*; *Dohrmann et al., 1996*; *Wall et al., 2000*; *Birsoy et al., 2006*; *Chen et al., 2006*; *Andersson et al., 2007*) and L-R patterning (*Hyatt et al., 1996*; *Hanafusa et al., 2000*; *Rankin et al., 2000*; *Wall et al., 2000*; *Peterson et al., 2013*). The *vg1* transcript was initially identified as being enriched in the vegetal cortex of *Xenopus* oocytes (*Rebagliati et al., 1985*) and was subsequently found to be able to induce mesoderm in animal cap assays (*Thomsen and Melton, 1993*; *Kessler and Melton, 1995*). Vg1 physically interacts with Nodal (*Tanaka et al., 2007*; *Fuerer et al., 2014*) and utilizes the same Nodal signaling pathway components (*Cheng et al., 2003*). The mammalian and *Xenopus* Vg1 orthologs may inhibit Bmp signaling in *Xenopus* and cultured cells (*Birsoy et al., 2006*; *Levine and Brivanlou, 2006*). Chimeric versions of the zebrafish Vg1 ortholog Gdf3 also induce mesoderm in animal cap assays (*Dohrmann et al., 1996*; *Peterson et al., 2013*), while morpholino oligonucleotide (MO)-mediated knockdown has demonstrated a requirement for Gdf3 in L-R patterning in zebrafish (*Peterson et al., 2013*).

*gdf3* is both maternally and zygotically expressed in zebrafish (*Helde and Grunwald, 1993*; *Dohrmann et al., 1996*), but the potential role of the endogenous protein in early embryonic patterning events has not been examined. To address this issue and further investigate the roles of Gdf3 during embryogenesis, we generated zebrafish *gdf3* mutants using CRISPR/Cas9 genome editing. Analysis of mutants revealed that maternally-supplied Gdf3 is essential for robust Nodal signaling in the early embryo, being required to form head and trunk mesoderm and endoderm (mesendoderm). Utilizing *gdf3* mutants and MO knockdown, we confirm the role of Gdf3 in promoting *spaw* expression during L-R patterning and demonstrate a role for Gdf3 in the formation of the zebrafish L-R coordinator, Kupffer's vesicle (KV). Taken together, our work supports a model of Gdf3 as an essential co-ligand for robust Nodal signaling throughout zebrafish development and, therefore, as an essential factor in vertebrate patterning.

## Results

### Maternal *gdf3* is required for embryonic mesendoderm patterning

*gdf3* is expressed maternally and ubiquitously during early stages of zebrafish development (*Figure 1A*) (*Helde and Grunwald, 1993*). The amount of *gdf3* mRNA diminishes rapidly through the blastula and gastrula stages and disappears by approximately 90% epiboly (*Helde and Grunwald, 1993*). This is followed by the return of expression in a tissue-restricted manner starting at the late bud stage and progressing into somitogenesis, when expression is observed in the lateral plate mesoderm (LPM) and in cells around KV (*Figure 1A*) (*Peterson et al., 2013*). This temporal and positional variation suggests that *gdf3* has multiple roles in zebrafish development.

To investigate these potential roles, we used CRISPR/Cas9 to generate stable zebrafish lines containing mutations in the first exon of the *gdf3* locus (*Figure 1B*, *Figure 1—figure supplement 1*). Each of the three mutant alleles contained a premature stop codon in the first exon, prior to the mature ligand domain-encoding region, and are thus predicted to be null alleles. Homozygous *gdf3* zygotic mutant embryos (Z*gdf3*) displayed no gross morphological phenotypes, were indistinguishable from heterozygous controls (*Figure 1C and D*), and grew to adulthood. By contrast, embryos lacking maternal *gdf3* (M*gdf3*) displayed overt developmental defects (*Figure 1E*) and did not survive beyond 3 days post fertilization (dpf). M*gdf3* and maternal-zygotic *gdf3* (MZ*gdf3*) embryos were indistinguishable from each other (*Figure 1E and F*), suggesting that the loss of Z*gdf3* did not contribute further to the gross M*gdf3* phenotype. Moreover, these defects could be rescued with wild-type (WT) *gdf3* mRNA (*Figure 1G and H*), a result that confirms the mutant phenotypes arose from disruption of the *gdf3* locus.

The defects in M*gdf3* and in MZ*gdf3* embryos strongly resemble the phenotypes exhibited by double mutants for the zebrafish Nodal orthologs *ndr1* and *ndr2* (*Feldman et al., 1998*) and MZ mutants for the essential Nodal co-receptor, *teratocarcinoma-derived growth factor 1* (*tdgf1*; previously known as *one eyed pinhead*) (*Gritsman et al., 1999*). Similarities include the dorsal accumulation of cells following convergence and epiboly in M*gdf3* and MZ*gdf3* embryos at the tailbud stage,

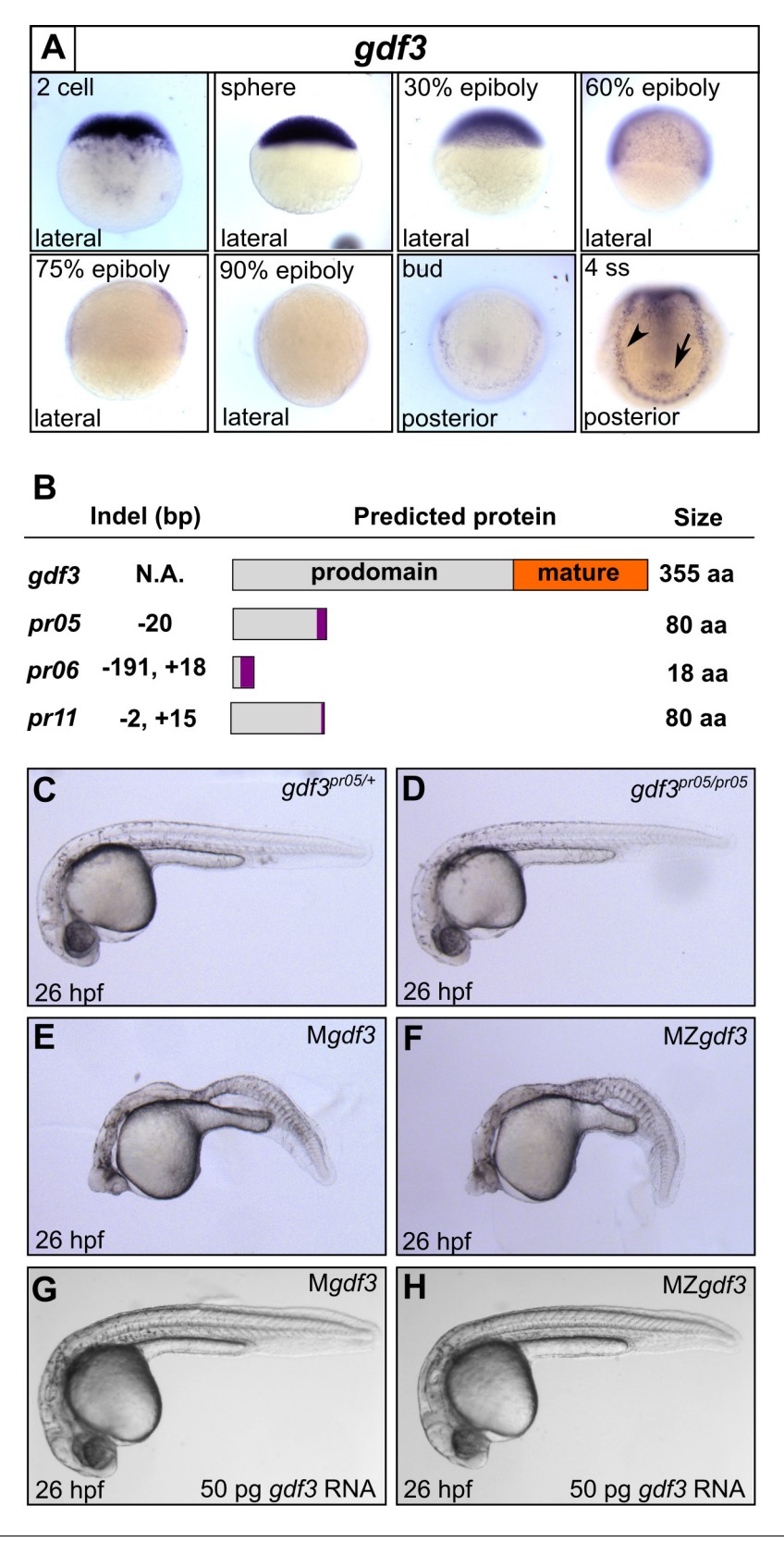

**Figure 1.** Mutations in maternal *gdf3* lead to defects in embryonic patterning. (**A**) RNA *in situ* hybridization shows *gdf3* mRNA is broadly expressed throughout early zebrafish development, followed by specific expression in *Figure 1 continued on next page*

*Figure 1 continued*

Kupffer's vesicle (arrow) and the lateral plate mesoderm (arrowhead). Views are indicated. (B) Schematic of *gdf3* mutant alleles predicted to form truncated proteins due to early stop codons in the first exon. Purple regions indicate changes in amino acid sequence prior to the premature stop codon in each allele (see Material and Methods for more details). (C–F) Loss of maternally deposited Gdf3 causes patterning defects. Embryos heterozygous (C) or homozygous (D) for the *pr05* allele appeared normal at 26 hpf. Heterozygous embryos from homozygous mothers (M*gdf*) exhibited defects in mesoderm and endoderm formation (E). Maternal-zygotic *gdf3* embryos (MZ*gdf3*) had similar mesoderm and endoderm defects as M*gdf3* embryos (F). *gdf3* mRNA injected at the one cell stage rescued defects in both M*gdf3* (G) and MZ*gdf3* (H) embryos. C-H are lateral views.
DOI: https://doi.org/10.7554/eLife.28635.002

The following figure supplements are available for figure 1:

**Figure supplement 1.** Schematic of *gdf3* mutant allele sequences.
DOI: https://doi.org/10.7554/eLife.28635.003
**Figure supplement 2.** Multiple *gdf3* mutant alleles have similar phenotypes.
DOI: https://doi.org/10.7554/eLife.28635.004

displacing the anterior most region of these embryos vegetally by 90 degrees compared to WT embryos (*Figure 1—figure supplement 2E and F*; compare to embryos in *Gritsman et al., 1999*). Additionally, M*gdf3* and MZ*gdf3* embryos at 26 hours post fertilization (hpf) are cyclopic and exhibit extensive defects in mesoderm and endoderm formation (*Figure 1E and F* and below), again reminiscent of *ndr1;ndr2* double mutants and MZ*tdgf1* mutants (*Feldman et al., 1998*; *Gritsman et al., 1999*). Thus, maternally-supplied *gdf3* is required for early embryogenesis, and loss of function cause defects that are consistent with strong reduction of early Nodal signaling.

## Gdf3 is necessary for the expression of Nodal target genes responsible for mesendoderm formation

Since M*gdf3* and MZ*gdf3* mutants resemble embryos lacking Nodal signaling, we assessed Nodal target gene expression in these mutants. The Nodal target gene *lefty1* (*lft1*) was absent in M*gdf3* and MZ*gdf3* embryos (*Figure 2A–C*). While expression of *ndr2* initiated in M*gdf3* and MZ*gdf3* embryos, it was expressed at lower levels compared with controls (*Figure 2D–F*); since Nodal signals propagate *ndr2* expression, this result and the absence of *lft1* suggests that Nodal signaling is markedly reduced or lost in the absence of maternal *gdf3*. In agreement, Nodal targets required for mesoderm and endoderm formation (*Feldman et al., 1998*) were also lost or decreased in M*gdf3* and MZ*gdf3* mutants. *goosecoid* (*gsc*) (*Stachel et al., 1993*), a marker of progenitors of the prechordal plate, is absent in M*gdf3* and MZ*gdf3* embryos (*Figure 2G–I*). Expression of *floating head* (*flh*) (*Talbot et al., 1995*), a Nodal target that marks notochord progenitors (*Gritsman et al., 2000*), is decreased in M*gdf3* and MZ*gdf3* embryos (*Figure 2J and K*).

Similarly, expression of *no tail* (*ntl*) in the dorsal region (*Schulte-Merker et al., 1992*), a domain that requires the highest levels of Nodal signaling (*Gritsman et al., 2000*), is absent in M*gdf3* and MZ*gdf3* embryos (*Figure 2L and M*). *axial* (*axl*) (*Strähle et al., 1993*), a marker of progenitors of axial mesoderm and endoderm, and *sox17* (*Alexander and Stainier, 1999*), a marker of presumptive endoderm, were both absent in M*gdf3* and MZ*gdf3* embryos (*Figure 2N–Q*). Taken together, these results suggest that Nodal signaling is strongly diminished or absent in M*gdf3* and MZ*gdf3* mutants and, as a result, mesoderm and endoderm are not induced. To confirm that these effects on Nodal target genes were a result of the loss of Gdf3, we injected *gdf3* mRNA into MZ*gdf3* embryos at the one-cell stage. This led to the recovery of *gsc* (*Figure 2R and S*), *lft1*, *ntl*, and *sox17* expression, (*Figure 2—figure supplement 1A–F*) confirming the role of *gdf3* in regulating Nodal target genes.

## Bmp and residual Nodal signaling promote tail mesoderm formation in M*gdf3* and MZ*gdf3* embryos

While M*gdf3* and MZ*gdf3* mutants strongly resemble *ndr1:ndr2* double mutants, M*gdf3* and MZ*gdf3* embryos consistently exhibit longer tails. This phenotype resembles that of MZ*foxh1*^*pr1/pr1* mutants in which a subset of Nodal signaling is eliminated by loss of the Foxh1 transcription factor, but signaling through other Nodal-activated transcription factors is retained (*Slagle et al., 2011*). To determine if low levels of Nodal signaling remain in the absence of maternal Gdf3, we injected M*gdf3*

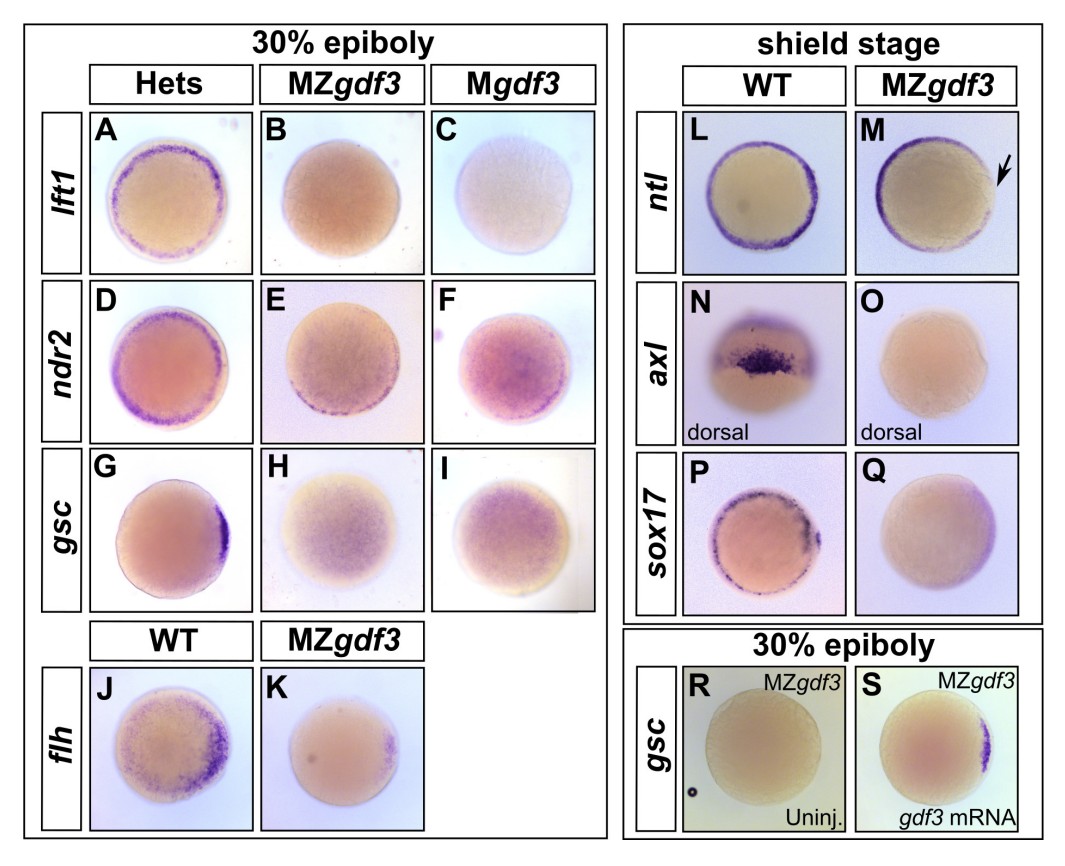

**Figure 2.** Gdf3 is necessary for the expression of Nodal target genes. (A–K) RNA *in situ* hybridization of Nodal target gene expression at 30% epiboly. (A–C) *lefty1* (*lft1*) is normally expressed in the margin (A), but is lost in maternal-zygotic *gdf3* mutant embryos (MZ*gdf3*, B) and heterozygous embryos derived from maternal homozygotes (M*gdf3*, C). (D–F) *cyclops* (*cyc*) is normally expressed in the margin (D). Expression of *cyc* is reduced in MZ*gdf3* (E) and M*gdf3* (F) embryos. (G–I) *goosecoid* (*gsc*) is normally expressed in the dorsal region (G), but is absent in MZ*gdf3* (H) and M*gdf3* (I) embryos. (J–K) *floating head* (*flh*) is normally expressed in the dorsal region (J) but is diminished in MZ*gdf3* (K) embryos. (L–Q) RNA *in situ* hybridization of Nodal target gene expression at the shield stage. (L–M) *no tail* (*ntla*) is expressed along the margin in WT embryos (L), but is absent from the dorsal end of MZ*gdf3* embryos (M, arrow). (N, O) *axial* (*axl*) is expressed in the dorsal region of WT embryos (N) but absent in MZ*gdf3* embryos (O). (P, Q) *sox17* is expressed in the margin of WT embryos (P) but absent in MZ*gdf3* embryos (Q). (R–S) Loss of *gsc* in MZ*gdf3* embryos (R) can be rescued by injection of 50 pg *gdf3* mRNA. All views are animal, with dorsal to the right and ventral to the left, unless otherwise indicated.

DOI: https://doi.org/10.7554/eLife.28635.005

The following figure supplement is available for figure 2:

**Figure supplement 1.** Loss of Nodal target gene expression can be rescued with *gdf3* mRNA injection.
DOI: https://doi.org/10.7554/eLife.28635.006

and MZ*gdf3* embryos with *ndr1:ndr2* MOs. Injection of the *ndr1:ndr2* MOs into MZ*gdf3* mutants led to a tail defect that closely resembled the tail of *ndr1:ndr2* double morphants (*Figure 3A and B*) and *ndr1:ndr2* double mutants (*Feldman et al., 1998*). We then overexpressed Nodal using exogenous *ndr1* mRNA to examine whether higher amounts of Nodal could activate target gene expression in MZ*gdf3* mutants. Although 1 pg of *ndr1* led to a modest expansion of *gsc* expression in WT embryos (*Figure 3C and D*), it did not activate *gsc* expression in MZ*gdf3* embryos (*Figure 3F and G*). By contrast, 10 pg of injected *ndr1* mRNA led to broad *gsc* expression in both WT and MZ*gdf3* mutant embryos (*Figure 3E and H*). These data, combined with our marker gene expression analysis, suggests that there is residual low-level Nodal activity present in the M*gdf3* and MZ*gdf3* mutant embryos that is inadequate for the formation of head and trunk endoderm.

Low levels of Nodal signaling act in concert with Bmp signaling to correctly form the posterior structures in the zebrafish embryo (*Fauny et al., 2009*). It has been reported that Gdf3 actively inhibits Bmp signaling in *Xenopus* and in cultured stem cells (*Levine and Brivanlou, 2006*). Additionally,

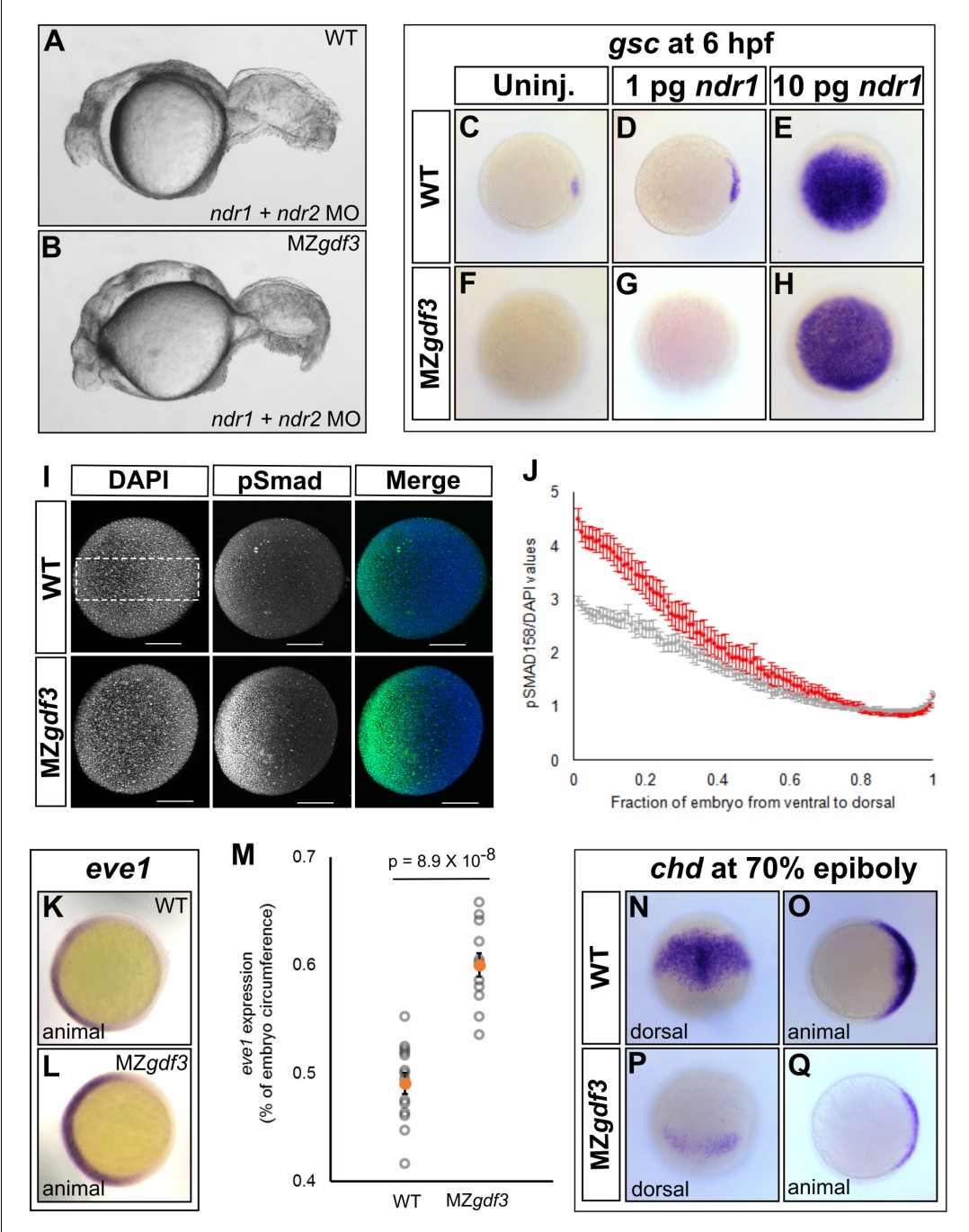

**Figure 3.** Loss of Gdf3 leads to expansion of Bmp signaling. (A) Knockdown of *nodal related 1* (*ndr1*) and *nodal related 2* (*ndr2*) in wild-type (WT) embryos causes complete loss of head and trunk mesendoderm and defects in tail patterning. (B) *ndr1:ndr2* knockdown in MZ*gdf3* embryos causes loss of residual tail patterning, producing embryos that more closely resemble those completely lacking Nodal signaling (A). (C-E) Overexpression of 1 pg or 10 pg *ndr1* led to increased expression of the Nodal target gene goosecoid (*gsc*) in WT embryos. (F-H) Overexpression of 10 pg *ndr1*, but not 1 pg, produced widespread expression of *gsc* in MZ*gdf3* mutants. (I) Immunostaining of phosphorylated Smad1/5/8 (green) counterstained with DAPI (blue). Ventral pSmad levels were increased in MZ*gdf3* mutants. (J) Plotting the normalized pSmad1/5/8 intensity over position on the ventral-dorsal axis confirms that MZ*gdf3* embryos (n = 14) have increased BMP signaling in the ventral region compared to WT (n = 13). (K-M) RNA *in situ* hybridization of *eve1* showed expansion of expression in the ventral region of MZ*gdf3* embryos (K) compared to WT (L). (M) Quantitative comparison of WT (n = 14) and MZ*gdf3* (n = 12) embryos confirmed expansion of the ventral *eve1* expression domain in MZ*gdf3* embryos. (N-Q) RNA *in situ* hybridization of *chordin* (*chd*) revealed its expression is reduced in MZ*gdf3* embryos. All views are animal unless otherwise indicated. In M, *p*-value obtained by Student's *t*-test (two-sided, homoscedastic). Error bars in J and M, standard error of the mean. Matlab code and data are available as *Figure 3—source code 1*, *2*, and *3* and *Figure 3—source data 1* and *2*.

*Figure 3 continued on next page*

*Figure 3 continued*

DOI: https://doi.org/10.7554/eLife.28635.007

The following source data and soruce codes are available for figure 3:

**Source code 1.** MATLAB script for quantification of the *eve1* RNA *in situ* stain.

DOI: https://doi.org/10.7554/eLife.28635.008

**Source code 2.** MATLAB script for quantification of phospho-SMAD1/5/8 immunostain in MZ*gdf3* mutant embryos.

DOI: https://doi.org/10.7554/eLife.28635.009

**Source code 3.** MATLAB script for quantification of phospho-SMAD1/5/8 immunostain in WT embryos.

DOI: https://doi.org/10.7554/eLife.28635.010

**Source data 1.** Raw data for quantification of the *eve1* RNA *in situ* stain in WT and MZ*gdf3* embryos.

DOI: https://doi.org/10.7554/eLife.28635.011

**Source data 2.** Raw data for quantification of the pSMAD1/5/8 immunostain in WT and MZ*gdf3* embryos.

DOI: https://doi.org/10.7554/eLife.28635.012

the depletion of Gdf3 in *Xenopus* results in reduced expression of the Bmp inhibitor *chordin* (*Birsoy et al., 2006*). We therefore examined Bmp signaling in M*gdf3* and MZ*gdf3* embryos by assessing levels of phosphorylated Smad1/5/8 (p-Smad1/5/8), a modification induced by Bmp signaling. At 70% epiboly, a ventral-to-dorsal increase in p-Smad1/5/8 was observed in MZ*gdf3* mutants compared to controls, with the ventral end exhibiting the largest increase in phosphorylation (*Figure 3I and J*). Consequently, we observed an expansion in expression of the Bmp target and ventral mesoderm marker *eve1* (*Joly et al., 1993*) towards the dorsal side of the embryo (*Figure 3K–M*). These data suggest that Gdf3 plays a role in modulating Bmp signaling and restricting the expansion of the ventral mesoderm. We also note that dorsal expression of the Bmp inhibitor *chordin* (*chd*) (*Figure 3N–Q*) is decreased in M*gdf3* and MZ*gdf3* embryos; this would also allow for the increased Bmp signaling observed in M*gdf3* and MZ*gdf3* embryos.

## Gdf3 is required for proper KV architecture and the expression of LPM *spaw*

Nodal signaling has at least two roles in L-R patterning at different stages of zebrafish development: first at an early stage by directing correct patterning of the dorsal cells to properly form KV (*Aamar and Dawid, 2010*; *Compagnon et al., 2014*), and second, by inducing expression of the *nodal* ortholog *spaw* in the LPM (*Long et al., 2003*; *Peterson et al., 2013*). Gdf3 is also required for Nodal signaling in the LPM, since MO knockdown of Gdf3 reduces the expression of the *nodal* ortholog *spaw* in this tissue (*Figure 4A*) (*Peterson et al., 2013*). Therefore, we expected L-R patterning defects in our mutants. However, Z*gdf3* mutants did not exhibit defects in the laterality of heart jogging (data not shown), an asymmetric morphogenetic event that acts as a measure of L-R patterning (*Baker et al., 2008*; *Smith et al., 2008*; *de Campos-Baptista et al., 2008*). Gdf3 is a stable protein since it remains detectable in the zebrafish embryo as late as 2.5 dpf, well beyond stages of reported mRNA expression (*Helde and Grunwald, 1993*; *Dohrmann et al., 1996*). In the absence of zygotic Gdf3, maternally supplied Gdf3 might compensate in L-R patterning events. Unfortunately, M*gdf3* and MZ*gdf3* mutants cannot be assessed for L-R patterning defects as they do not produce LPM or mesendoderm-derived organs. Therefore, we used a reported translation-blocking MO (*Ye et al., 2010*) to reduce the amount of maternal and zygotic Gdf3 to levels that allow for mesendoderm formation, and yet allow us to further analyze the role of Gdf3 in L-R patterning.

KV development is aberrant in MO-injected embryos. Normally, the cells in the anterior-dorsal region of KV are columnar and more tightly packed as compared to the ventral region (*Figure 4B*, see uninjected). This architecture is proposed to generate left-ward asymmetric flow that is faster across the anterior-dorsal portion of KV (*Prabhat et al., 2007*; *Okabe et al., 2008*; *Supatto et al., 2008*; *Wang et al., 2012*; *Grimes and Burdine, 2017*). In Gdf3 knockdowns, this architecture is lost and all the cells exhibited a cuboidal morphology (*Figure 4B*, see *gdf3* MO). These data demonstrate that Gdf3 knockdown can adversely affect the formation of KV, which would prevent the generation of normal asymmetric fluid flow.

Proper KV formation and flow is important for generating asymmetric expression of the Nodal inhibitor *dand5* in KV (*Hashimoto et al., 2004*; *Hojo et al., 2007*; *Schweickert et al., 2010*;

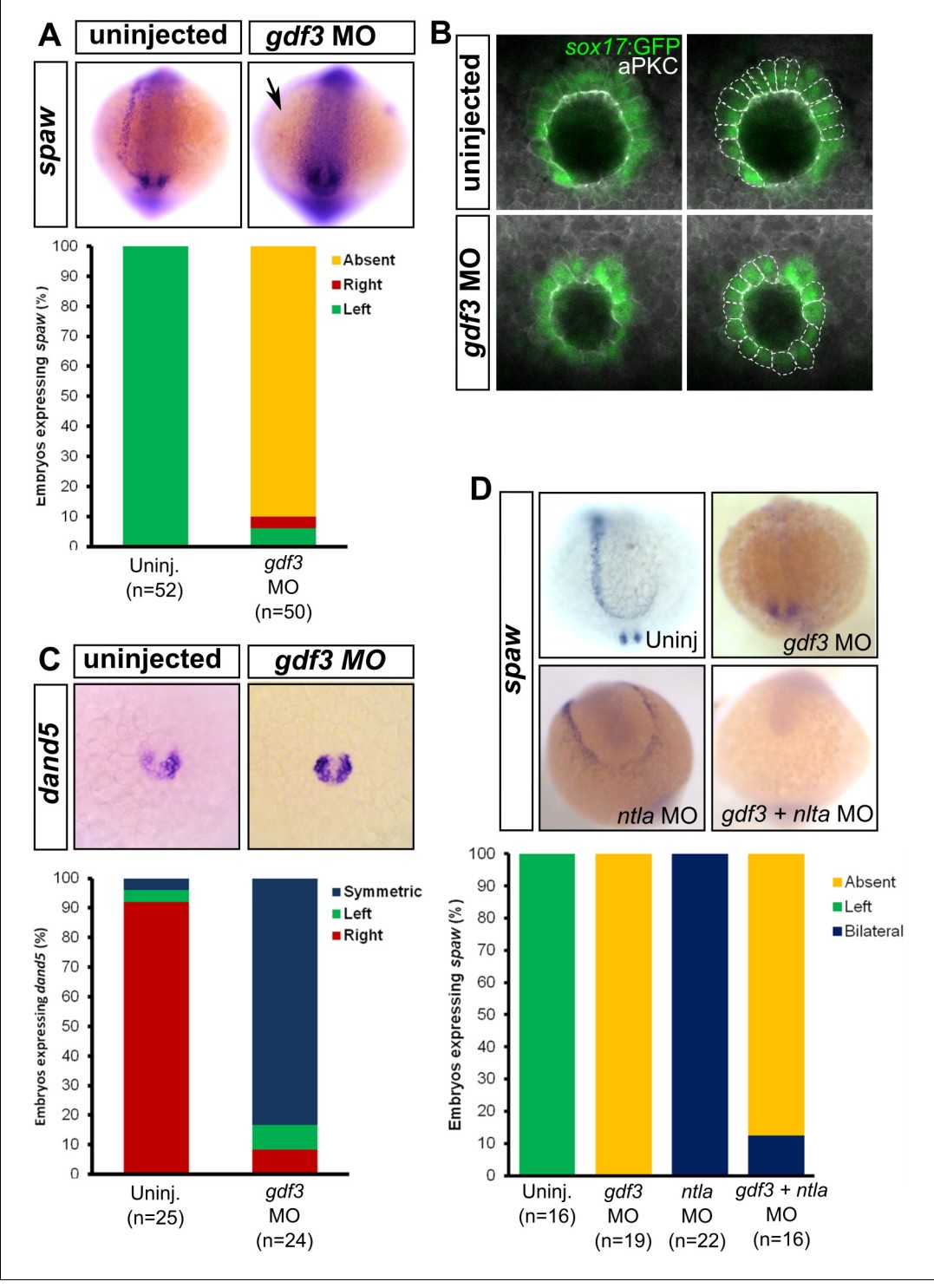

**Figure 4.** Attenuated production of Gdf3 leads to left-right patterning defects. (**A**) *spaw* expression is altered in *gdf3* morphants, with most lacking expression in the lateral plate mesoderm (arrow), as shown by RNA *in situ* hybridization. (**B**) Immunostaining of aPKC (white) and GFP driven by the *sox17* regulatory region (green) in Kupffer's Vesicle (KV) at 10 ss. The anterior-posterior asymmetry in cell morphology is lost in *gdf3* morphants. Cells are outlined by white dashed lines. (**C**) RNA *in situ* hybridization reveals *dand5* expression is symmetric around KV in *gdf3* morphants at 10 ss. (**D**) Loss of *ntla* leads to robust *spaw* expression in both the left and right LPM and a loss of KV and associated gene expression. *spaw* expression in *ntla + gdf3* double morphants is completely lost,
*Figure 4 continued on next page*

*Figure 4 continued*
supporting the idea that Gdf3 is required for robust Nodal expression given Nodal is unable to activate in the absence of inhibitors. Raw data provided in *Figure 4—source data 1–3*.
DOI: https://doi.org/10.7554/eLife.28635.013
The following source data is available for figure 4:
**Source data 1.** Raw data used to generate the *spaw* RNA *in situ* results bar graph in *Figure 4A*.
DOI: https://doi.org/10.7554/eLife.28635.014
**Source data 2.** Raw data used to generate the *dand5* RNA *in situ* results bar graph in *Figure 4C*.
DOI: https://doi.org/10.7554/eLife.28635.015
**Source data 3.** Raw data used to generate the *spaw* RNA *in situ* results bar graph in *Figure 4D*.
DOI: https://doi.org/10.7554/eLife.28635.016

*Lopes et al., 2010*). In contrast to WT embryos which exhibit a right-biased expression of *dand5*, Gdf3 knockdowns expressed *dand5* symmetrically (*Figure 4C*). To determine if the symmetrical expression of *dand5* leads to the loss of *spaw* expression in the left LPM of Gdf3 knockdowns, we prevented both the formation of KV and the expression of *dand5* by also knocking down *ntla* with a MO. Loss of *ntla* leads to loss of KV, *dand5* expression, and notochord (*Amack et al., 2007*; *Hashimoto et al., 2004*; *Odenthal et al., 1996*). In *ntla* mutants and morphants, *spaw* expression occurs at high levels in both the left and right LPM, presumably due to the loss of *dand5* at KV and the Nodal inhibitor *lefty1* in the midline (*Long et al., 2003*; *Lenhart et al., 2011*; *Burdine and Grimes, 2016*) (*Figure 4D*). We found that coinjection of *ntla* MO did not restore *spaw* expression in the LPM of *gdf3* morphants (*Figure 4D*). This suggests that Gdf3 is required for robust Spaw signaling, and without Gdf3, Spaw is unable to induce its own expression, even in the absence of midline-derived Spaw inhibitors.

## Rescue of mesendoderm defects in MZ*gdf3* embryos reveals a role for zygotic Gdf3 in L-R patterning

To confirm our Gdf3 knockdown results, and to further examine the potential role of zygotic *gdf3* in L-R patterning, we overexpressed *gdf3* mRNA to rescue the early mesendoderm defects in M*gdf3* and MZ*gdf3* embryos, and then analyzed L-R asymmetry in the embryos. A similar strategy was utilized to examine L-R patterning in MZ*tdgf1* embryos (*Yan et al., 1999*). This approach additionally allows us to examine potential differences in L-R patterning between M*gdf3* and MZ*gdf3* mutants, i.e. in the presence or absence of zygotic *gdf3*, respectively. Injection of 50 pg of *gdf3* mRNA in M*gdf3* embryos, which retain zygotic expression of *gdf3*, rescued mesendoderm formation and resulted in ~90% of embryos with normal L-R patterning based on the direction of heart jogging (*Figure 5A*). Injection of 50 pg of *gdf3* mRNA in MZ*gdf3* embryos also rescued mesendoderm defects but these embryos exhibited defects in heart jogging laterality, with ~50% of hearts jogging incorrectly to the right or remaining midline (*Figure 5A*). These data suggest that zygotic *gdf3* facilitates left-right patterning in the absence of sufficient maternal *gdf3* contribution. However, since Z*gdf3* mutants exhibit no L-R defects, zygotic Gdf3 is not required for L-R patterning in the presence of maternal Gdf3. In further support of the hypothesis that Gdf3 produced early in development is stable and sufficient to function in L-R patterning, injecting increasing amounts of *gdf3* mRNA increased the number of MZ*gdf3* embryos with correct leftward jogged hearts (*Figure 5A*).

We next analyzed molecular markers of L-R asymmetry in rescued mutants. While the LPM does not form in M*gdf3* and MZ*gdf3* embryos, presence of LPM and expression of *spaw* in the LPM is rescued in mutant embryos injected with *gdf3* mRNA (*Figure 5B*). In injected MZ*gdf3* embryos, LPM *spaw* laterality was irregular with ~40% of embryos displaying correct left-sided expression at the 12 somite stage (ss). In injected M*gdf3* embryos, where zygotic *gdf3* is still present, the number of embryos exhibiting correct left-sided expression of *spaw* increased to ~70%. This further supports the conclusion that zygotic *gdf3* contributes to correct L-R patterning in the absence of maternal Gdf3.

Since embryos treated with *gdf3* MO exhibited defects in KV cell morphology and functionality, we analyzed KV functionality in mesendoderm-rescued M*gdf3* and MZ*gdf3* embryos. While correct asymmetric *dand5* expression in KV was restored in rescued M*gdf3* embryos (*Figure 5C and H–I*),

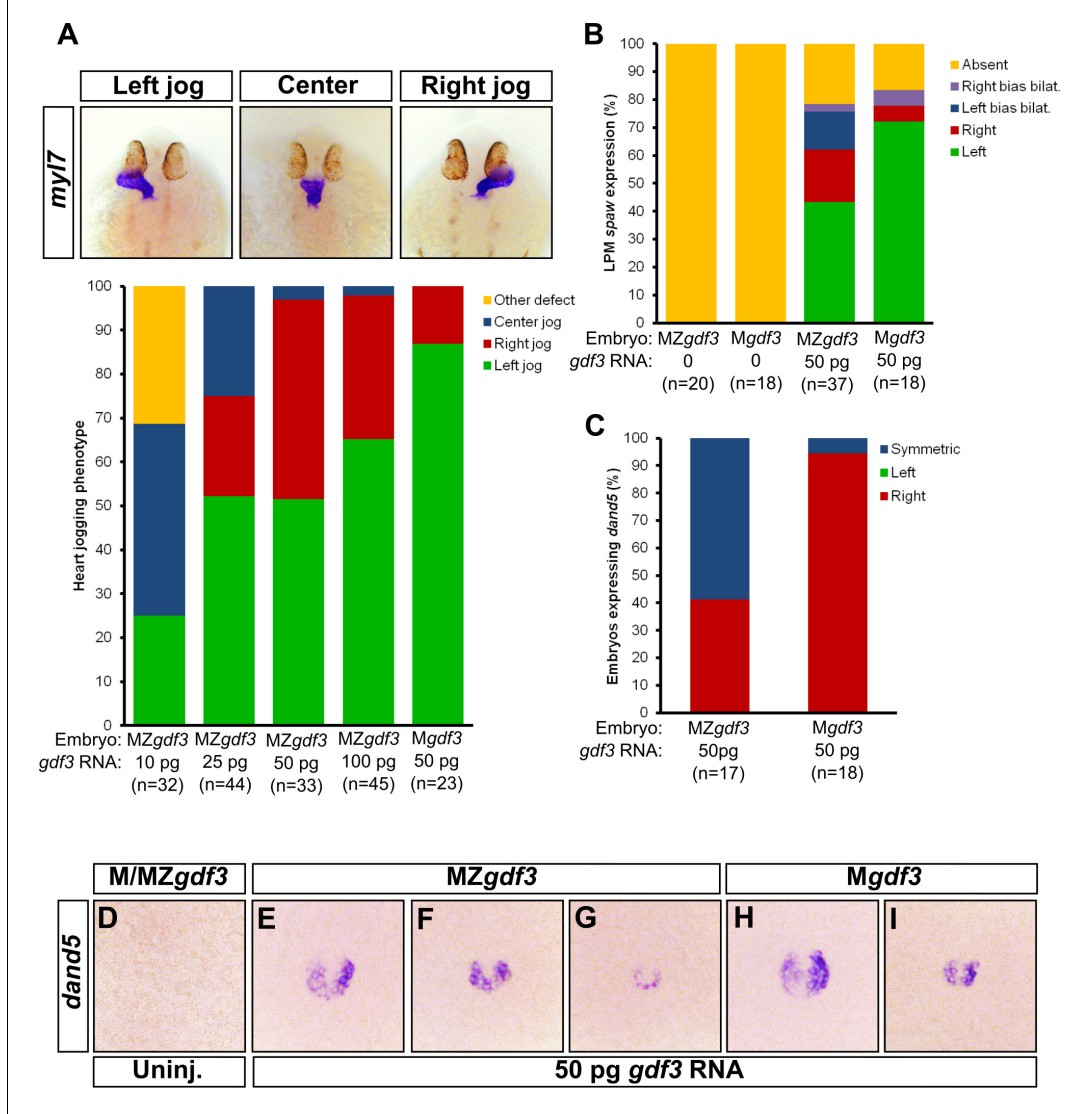

**Figure 5.** Zygotic Gdf3 can restore proper left-right patterning to embryos rescued from early mesendoderm defects. (A) Heart jogging defects improved with increasing amounts of injected *gdf3* mRNA in MZ*gdf3* embryos. Zygotic *gdf3* (present in M*gdf3* embryos) further rescued heart jogging defects in embryos injected with 50 pg of *gdf3* mRNA. (B) M/MZ*gdf3* embryos lack *spaw* expression at 12 ss. Embryos injected with 50 pg of *gdf3* mRNA rescued *spaw* expression in the LPM. Proper *spaw* expression in the left LPM was further restored in the presence of zygotic *gdf3*. (C) *dand5* expression was recovered in *MZgdf3* embryos injected with *gdf3* mRNA, with most embryos containing symmetric expression of *dand5* in KV (F,G). (C) Most M*gdf3* embryos injected with *gdf3* mRNA contained proper right-biased asymmetric expression of *dand5* (H,I). (D) *dand5* expression was absent in all uninjected M/MZ*gdf3* embryos. Raw data provided in *Figure 5—source data 1–3*.

DOI: https://doi.org/10.7554/eLife.28635.017

The following source data is available for figure 5:

**Source data 1.** Raw data used to generate the bar graph of heart jog results in *Figure 5A*.
DOI: https://doi.org/10.7554/eLife.28635.018

**Source data 2.** Raw data used to generate the *spaw* RNA *in situ* results bar graph in *Figure 5B*.
DOI: https://doi.org/10.7554/eLife.28635.019

**Source data 3.** Raw data used to generate the *dand5* RNA *in situ* results bar graph in *Figure 5C*.
DOI: https://doi.org/10.7554/eLife.28635.020

rescued MZ*gdf3* embryos expressed *dand5* symmetrically (*Figure 5C and F–G*), indicating loss of functional flow in KV, likely owing to aberrant KV morphology. These data suggest that zygotic Gdf3 is required for proper formation and function of KV in the absence of maternal Gdf3.

## Discussion

Signaling by multiple members of the TGFβ superfamily of ligands is important for patterning of early vertebrate embryos. Here, we show that the gene encoding the Vg1 ortholog in zebrafish, *gdf3*, is a maternal effect gene for early embryonic patterning. We provide evidence that the role of maternally supplied Gdf3 is to promote robust levels of Nodal signaling. Others have reported that Nodal and Vg1 orthologs can bind to one another (*Tanaka et al., 2007*; *Fuerer et al., 2014*). Furthermore, Vg1 requires the same pathway components as Nodal ligands for signaling, including the essential co-receptor Tdgf1 (*Cheng et al., 2003*). Thus, we hypothesize that Gdf3 dimerizes with Nodal to facilitate Nodal signaling during development. Our evidence suggests that lower levels of Nodal signaling still occur in the absence of maternal Gdf3. For instance, M*gdf3* and MZ*gdf3* embryos produce more posterior tail structures than embryos completely lacking Nodal signaling. Although Nodal can signal independently of Gdf3, target genes required for head and trunk mesendoderm are not activated without Gdf3 unless Nodal is overexpressed beyond physiological levels. Moreover, we find correct spatial expression of mesendoderm marker genes in MZ*gdf3* embryos injected with *gdf3* mRNA, which suggests that Gdf3 plays a permissive role in mesendoderm formation while spatial restriction of *ndr1* and *ndr2* expression is critical for patterning. Gdf3 has also been proposed to act as a direct inhibitor of Bmp signaling (*Birsoy et al., 2006*; *Levine and Brivanlou, 2006*), and we observe an increase in Bmp signaling in M*gdf3* and MZ*gdf3* embryos by assaying for pSMAD1/5/8. However, we see reduction of Bmp antagonists in these embryos, such as *chordin*, and thus additional studies are warranted to determine if Gdf3 is playing a direct or indirect role in modulating Bmp signaling in the early embryo.

We demonstrate that knockdown of Gdf3 can affect L-R patterning at two crucial steps: in formation of the L-R coordinator KV and in promoting robust expression of Nodal/Spaw in the LPM. For KV to form properly, correct expression of *sox17* is required (*Aamar and Dawid, 2010*), which itself requires robust Nodal signaling during germ layer formation (*Sako et al., 2016*). Furthermore, regional cell shape differences in KV depend on the presence of the neighboring notochord (*Compagnon et al., 2014*) which is absent in Nodal-signaling deficient embryos (*Feldman et al., 1998*). In our Gdf3 knockdown experiments, KV fails to acquire the regional cell shape differences that promote correct fluid flow, which is required for proper asymmetric expression of *dand5* (*Figure 6C and D*) (*Schweickert et al., 2010*; *Lopes et al., 2010*). In addition to this phenotype, knockdown of Gdf3 also prevents *spaw* expression from occurring in the LPM even in the absence of Nodal inhibitors Dand5 and Lefty1 (*Figure 6E*). These knockdown experiments suggest that Gdf3 plays a role both in KV organization and downstream, in the activation of LPM *spaw*. A previous report suggested that Gdf3 knockodwn did not affect KV, and that the loss of Spaw expression in the LPM of these embryos could be rescued by injection of *dand5* or *lefty1* morpholinos (*Peterson et al., 2013*). However, given that our data suggest maternal Gdf3 is responsible for proper L-R patterning, we suggest the differences in our results reflect the amount by which maternal Gdf3 is reduced with the two different AUG morpholinos that were used.

Analysis of L-R patterning events in mesendoderm-rescued M*gdf3* and MZ*gdf3* complemented our Gdf3 knockdown experiments, and confirmed roles for Gdf3 in L-R patterning. Importantly, we demonstrated that the presence of zygotic *gdf3* improved the rescue of L-R patterning defects in mesendoderm-rescued mutants. The lack of L-R patterning phenotypes in Z*gdf3* mutants was unexpected, especially since *gdf3* is zygotically expressed in very strict domains that also express crucial L-R patterning genes such as the KV region and the LPM. It could be that while maternal Gdf3 is sufficient for proper L-R patterning, zygotic expression provides robustness to the system. In the conditions used to raise zebrafish in the laboratory, such as controlled temperature, abundant food, and lack of predators, maternally supplied *gdf3* is sufficient for this process. However, in situations where lower levels of maternal product are supplied, zygotic Gdf3 might then be required alongside maternal Gdf3 for robust L-R patterning.

Our results demonstrate that Gdf3 acts permissively to promote robust Nodal signaling, and yet is spatially restricted to KV and LPM during L-R patterning. Given that other L-R patterning

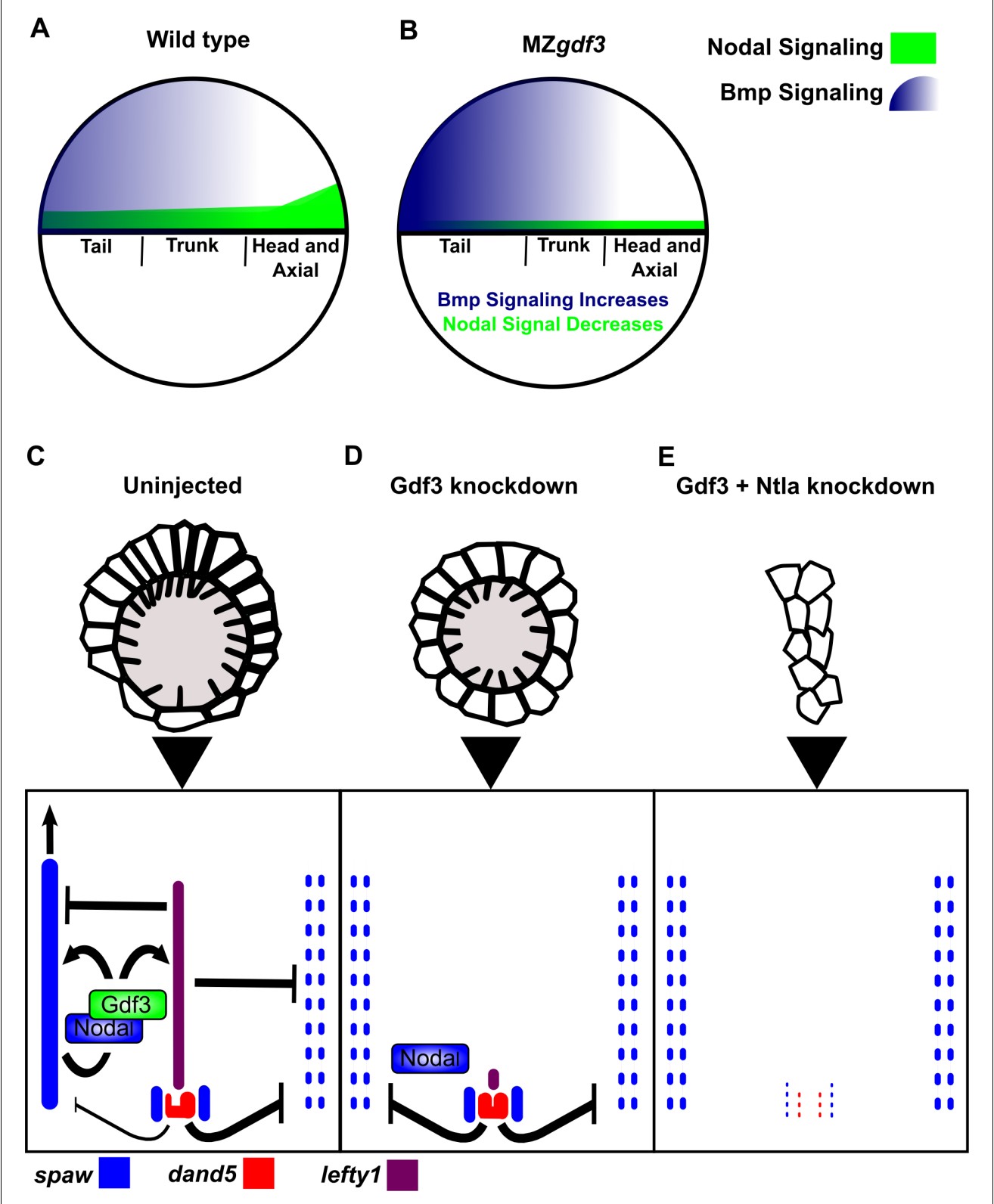

**Figure 6.** Model of Gdf3 function in early zebrafish development. (**A**) In wild-type (WT) embryos, Bmp signaling (blue) is established in a ventral-to-dorsal gradient during blastula and gastrula stages, while Nodal signaling (green) is established along the margin, with strongest expression on the dorsal side. (**B**) In MZ*gdf3* embryos, Nodal signaling is weak leading to reduced expression of Nodal target genes including the Bmp inhibitor Chordin. Bmp signaling is increased and leads to an extension of ventral mesoderm markers. However, as Nodal signaling is depleted or lost, the majority of

*Figure 6 continued on next page*

*Figure 6 continued*

mesoderm and endodermal cell fates fail to be maintained. (**C**) Proper development of Kupffer's vesicle includes the generation of asymmetries in cell shape along the anterior-posterior axis of the structure. Anterior cells, closest to the notochord, take on columnar shapes and become more tightly packed, leading to a greater number of cilia in this region. This architecture is required for asymmetric fluid flow in KV and proper *dand5* expression. (**D**) Attenuation of maternal Gdf3 leads to improper cell morphology during KV development, which in turn leads to irregular *dand5* expression. Dysregulation of dand5 expression on the left may then inhibit *spaw* in the left LPM. (**E**) Removal of Nodal antagonists Lefty1 and Dand5 via *ntla* morpholino-mediate knockdown does not restore *spaw* expression in the LPM. This suggests that Gdf3 plays a more direct role in regulating *spaw* expression, independent of its effects on KV development. Dotted lines (**C, D, E**) represent lost gene expression.

DOI: https://doi.org/10.7554/eLife.28635.021

components are expressed in these distinct domains at this time, *gdf3* expression could be restricted to these same domains by similar patterning mechanisms, even if spatial restriction of its expression is not required. Taken together, we propose that Gdf3 and Spaw dimerize to promote robust Nodal signaling in the LPM. In the future, it will be interesting to see if dimerization of these ligands allows Spaw to more efficiently travel to, and propagate through the LPM.

Collectively, our data suggest that Gdf3 functions in multiple developmental processes. During the blastula and gastrula stages, Gdf3 affects Nodal activity required for organization of the vertebrate embryo as well as the formation of KV. During somitogenesis it is required for Nodal signaling in the LPM. This agrees with work performed in mouse, where lines with targeted mutations in the Vg1 mouse orthologs Gdf1 and Gdf3 exhibited defects in mesoderm formation and L-R patterning (*Rankin et al., 2000*; *Andersson et al., 2006*; *2007*). Our work also suggests that Gdf3 functions as a co-ligand for Nodal signaling. Heterodimerization of other zebrafish TGFβ ligands in BMP signaling (*Little and Mullins, 2009*), lead us to propose a similar function for Gdf3 and Nodal ligands in Nodal signaling. Indeed, there is strong evidence that Vg1 orthologs can heterodimerize with Nodal to allow for robust or more efficient Nodal signaling (*Tanaka et al., 2007*; *Fuerer et al., 2014*). Overall, we conclude that Gdf3 facilitates Nodal ligands in generating a robust level of signaling activity required for both mesendoderm formation and proper L-R patterning.

# Materials and methods

**Key resources table**

| Reagent type (species) or resource | Designation | Source or reference | Identifiers | Additional information |
|---|---|---|---|---|
| gene (*Danio rerio*) | *growth differentiation factor 3 (gdf3)* | NA | ZFIN:ZDB-GENE-980526–389 | |
| gene (*Danio rerio*) | *nodal-related 1 (ndr1)* | NA | ZFIN:ZDB-GENE-990415–256 | |
| gene (*Danio rerio*) | *nodal-related 2 (ndr2)* | NA | ZFIN:ZDB-GENE-990415–181 | |
| gene (*Danio rerio*) | *southpaw (spaw)* | NA | ZFIN:ZDB-GENE-030219–1 | |
| gene (*Danio rerio*) | *DAN domain family, member 5 (dand5)* | NA | ZFIN:ZDB-GENE-040421–2 | |
| genetic reagent (*D. rerio*) | *gdf3 allele pr05* | this paper | | CRISPR/Cas9 generated deletion, predicted null |
| genetic reagent (*D. rerio*) | *gdf3 allele pr06* | this paper | | CRISPR/Cas9 generated indel, predicted null |
| genetic reagent (*ID. rerio*) | *gdf3 allele pr11* | this paper | | CRISPR/Cas9 generated indel, predicted null |
| antibody | Anti-Digoxigenin-AP | Roche, 11093274910 | | 1:3500 |
| antibody | Anti-phosphorylated SMAD1/5/8 | Cell Signaling Technologies, 9511S | | 1:100 |
| antibody | anti-PKCzeta (aPKC) | Santa Cruz Biotechnology, SC-216 | | 1:400 |
| antibody | Goat anti-Rabbit Alexa Fluor 647 | Invitrogen, A21246 | | 1:400 |

*Continued on next page*

Continued

| Reagent type (species) or resource | Designation | Source or reference | Identifiers | Additional information |
|---|---|---|---|---|
| antibody | Donkey anti-Rabbit Alexa Fluor 647 secondary antibody | Invitrogen, A31573 | | 1:400 |
| recombinant DNA reagent | pCS2-nCas9n | Addgene plasmid # 47929 | | |
| recombinant DNA reagent | pT7-gRNA | Addgene plasmid # 46759 | | |
| recombinant DNA reagent | pCS2(+)-gdf3-WT | this paper | | |
| recombinant DNA reagent | pCS2-3XFlag-sqt | PMID:27101364 | | |
| recombinant DNA reagent | gdf3 in situ probe plasmid | PMID: 8405668 | | |
| recombinant DNA reagent | ndr2 in situ probe plasmid | PMID: 9707578 | | |
| recombinant DNA reagent | flh in situ probe plasmid | PMID: 7477317 | | |
| recombinant DNA reagent | ntla in situ probe plasmid | PMID: 1295726 | | |
| recombinant DNA reagent | gsc in situ probe plasmid | PMID: 8104775 | | |
| recombinant DNA reagent | lft1 in situ probe plasmid | PMID: 10375514 | | |
| recombinant DNA reagent | axial in situ probe plasmid | PMID: 7687227 | | |
| recombinant DNA reagent | sox17 in situ probe plasmid | PMID: 10531029 | | |
| recombinant DNA reagent | spaw in situ probe plasmid | PMID: 12702646 | | |
| recombinant DNA reagent | dand5 in situ probe plasmid | PMID: 15084459 | | |
| sequence-based reagent | gdf3 morpholino | Genetools | | 5'-GCTCTGAGGAGGACCAAGAACATTA-3' |
| sequence-based reagent | ntla morpholino | Genetools | | 5'- GACTTGAGGCAGGCATATTTCCGAT-3' |
| sequence-based reagent | ndr1 morpholino | Genetools | | 5'-ATGTCAAATCAAGGTAATAATCCAC-3' |
| sequence-based reagent | ndr2 morpholino | Genetools | | 5'-GCGACTCCGAGCGTGTGCATGATG-3' |
| peptide, recombinant protein | T4 PNK | NEB, M0201S | | |
| peptide, recombinant protein | T4 DNA ligase | NEB, M0202S | | |
| peptide, recombinant protein | BsmBI | NEB, R0580S | | |
| peptide, recombinant protein | BglII | NEB, R0144S | | |
| peptide, recombinant protein | SalI | NEB, R0138S | | |
| peptide, recombinant protein | BamHI | NEB, R0136S | | |
| peptide, recombinant protein | XbaI | NEB, R0145S | | |

*Continued*

| Reagent type (species) or resource | Designation | Source or reference | Identifiers | Additional information |
|---|---|---|---|---|
| commercial assay or kit | Qiaquick PCR Purification Kit | Qiagen, 28104 | | |
| commercial assay or kit | QIAquick Gel Extraction Kit | Qiagen, 28704 | | |
| commercial assay or kit | mMessage mMachine SP6 | ThermoFisher, AM1340 | | |
| commercial assay or kit | MEGAshortscript T7 Transcription kit | ThermoFisher, AM1354 | | |
| commercial assay or kit | QIAprep Spin Miniprep kit | Qiagen, 27106 | | |
| commercial assay or kit | Phusion High-Fidelity PCR Kit | EB, E0553L | | |
| software, algorithm | Imaris | | | Image analysis and quantification |
| software, algorithm | ImageJ | | | Image analysis and quantification |
| software, algorithm | MATLAB | | | Image analysis and quantification |

## Fish care

Zebrafish husbandry was performed using protocols in accordance with Princeton University Institutional Animal Care and Use Committee. Embryos were maintained at 28°C in Egg Water (300 mg/L Instant Ocean Sea Salts and 0.02 mg/L Methylene Blue).

## Generation of *gdf3* mutants

Insertions and/or deletions were produced in the first exon of zebrafish *gdf3* using the CRISPR-*Cas9* genome editing system. Target sites GGGTACGAGGAAACATCGTG (Chr17:4086653–4086672; Ensembl 88:10; genome assembly GRCz10) and GGGTCAGAAGACAGGCTCTG (17:4086856–4086875 Ensembl 88:10; genome assembly GRCz10) were selected using CHOPCHOP (*Montague et al., 2014*) (*Figure 1—figure supplement 1*). The sgRNAs were made with the guidance of published protocols (*Cong et al., 2013*; *Jao et al., 2013*). Oligos for each target site were annealed (5'-TAGGGTACGAGGAAACATCGTG-3' annealed to 5'-AAACCACGATGTTTCCTCGTAC-3')(5'-TAGGGTCAGAAGACAGGCTCTG-3' annealed to 5'-AAACCAGAGCCTGTCTTCTGAC-3') and phosphorylated at the 5'-end by incubating a 10 µl reaction mixture (1X T4 ligase buffer (NEB, B0202S), 10 µM each of the two oligos from a target-site, and 5 units of T4 PNK (NEB, M0201S)) with the following heating profile: 30 min at 37°C, 5 min at 95°C, a 2 °C/s decrease in temperature until 85°C, and cooled at 0.1 °C/s until 25°C. The annealed oligos were cloned into pT7-gRNA (*Jao et al., 2013*) by incubating a 10 µl reaction mixture (40 ng/µl of pT7-gRNA, a 1:20 dilution of the annealed oligos for an individual target site, 1X NEB buffer 3.1 (NEB, B7203S), 1X T4 ligase buffer (NEB, B0202S), 5 units of *Bsm*BI (NEB, R0580S), 5 units of *Bgl*II (NEB, R0144S), 5 units of *Sal*I (NEB, R0138S), and 200 units of T4 DNA ligase (NEB, M0202S)) with the following heating profile: 3 cycles of 20 min at 37°C, 15 min at 16°C, 10 min at 37°C, and 15 min at 55°C. Plasmids with the correct insertion were confirmed by Sanger sequencing. Plasmids were transformed into *E.coli*. DH5α, grown, and isolated using a QIAprep Spin Miniprep kit (Qiagen, 27106). The sgRNAs were then synthesized from *Bam*HI (NEB, R0136S) digested plasmids using a MEGAshortscript T7 Transcription kit (ThermoFisher, AM1354). *Cas9* mRNA was synthesized from *Not*I (NEB, R0189S) digested pCS2-nCas9n (*Jao et al., 2013*) using an mMessage mMachine SP6 Transcription Kit (ThermoFisher, AM1340). Both pT7-gRNA (Addgene plasmid # 46759) and pCS2-nCas9n (Addgene plasmid # 47929) were gifts from Wenbiao Chen.

Embryos were injected into the single cell at the one cell stage with a ~1.8 nl RNA mixture consisting of 240 pg of Cas9 mRNA and 10 pg of each sgRNA. The embryos were raised and incrossed to check for mutant phenotypes. Candidates containing potential germline transmission of a *gdf3* mutant allele were isolated by outcrossing to WT zebrafish. Homozygous mutants for a *gdf3* mutant allele were generated by incrossing heterozygous fish for that allele. Compound heterozygous fish with two different mutant alleles were generated by crossing heterozygous fish containing different

mutant alleles. Mutations in the *gdf3* locus were confirmed using PCR genotyping. Tail fin-clips were added to 50 µl of Lysis Buffer (10 mM Tris-HCl at pH 8.0, 50 mM KCl, 0.3% Tween 20, 0.3% NP40, and 1 mM EDTA) and incubated at 98°C for 10 min. The samples received 11 µl of 10 mg/ml ProK and were incubated overnight at 55°C. The samples were then incubated at 98°C for 10 min, cooled on ice, and diluted 1:18 with nuclease free water. PCR was performed using 5 µl of the diluted DNA extract. Phusion High-Fidelity PCR Kit (NEB, E0553L) was used with forward primer 5'-CAGA TCCCAAAATCTTCATGC-3' and reverse primer 5'-AAGTGTGGTCCGTAAGATCCTC-3' to amplify the region containing potential mutations (Heating profile: 98°C for 30 s, followed by 30 cycles of 98°C for 10 s, 66°C for 30 s, and 72°C for 30 s, followed by 72°C for 5 min). DNA was size separated on a 3% Metaphor gel (Lonza, 50181) containing ethidium bromide and visualized using a UV light source. The bands were excised, gel purified using a Qiaquick Gel Extraction Kit (Qiagen, 28704), and sequenced.

Three mutations in *gdf3* were obtained (*Figure 1—figure supplement 1*). Allele pr05 contains a 20 base pair (bp) deletion beginning at base 218 of the coding region, which causes a frame shift in eight amino acids (aa) and leads to a premature stop codon. This allele is predicted to produce an 80 aa truncated version of Gdf3. Allele pr06 contains a 189 bp deletion beginning at base 21 of the coding region immediately followed by a 3 bp insertion, and a second lesion that consists of a 15 bp insertion at base 236 immediately followed by a 2 bp deletion. The first lesion causes a frameshift that changes the sequence of 11 aa prior to a premature stop codon which is predicted to produce an 18 aa truncated version of Gdf3. Allele pr11 contains a 15 bp insertion after 235 bases in the coding region immediately followed by a 2 bp deletion. This indel causes a framshift that changes two aa prior to a premature stop codon which is predicted to produce an 80 aa truncated version of Gdf3.

Since, Mgdf3 and MZgdf3 embryos exhibited the same phenotype, they were used interchangeably. Additionally, all three gdf3 mutant alleles caused the same M/MZgdf3 phenotypes and did not complement each other (*Figure 1—figure supplement 2A–D*, data not shown), which confirmed that the defects in embryonic patterning we observed were due to a dysfunctional *gdf3* locus. Since all the mutants exhibited the same phenotype, *gdf3* mutants containing the pr05 allele were used for most this work.

Images of gross morphological phenotypes were acquired using a Leica DFC365 FX camera attached to a Leica M205 FA stereomicroscope.

## Morpholino and RNA rescue injections

### Morpholino injections

The previously published *gdf3* AUG morpholino (*Ye et al., 2010*) (5'-GCTCTGAGGAGGACCAA-GAACATTA-3'; Genetools), was used to knockdown translation of Gdf3. Each injection contained 5 ng of *gdf3* AUG morpholino and 5 ng of *p53* AUG morpholino (used to limit off-target effects [*Ye et al., 2010*]). Approximately 2 ng of the previously published *ntla* AUG morpholino (*Nasevicius and Ekker, 2000*) (5'- GACTTGAGGCAGGCATATTTCCGAT-3'; Genetools), was used to knockdown translation of Ntla. A mixture containing 10 ng of *cyc* (*Karlen and Rebagliati, 2001*) and 10 ng of *sqt* (*Feldman and Stemple, 2001*) AUG morpholinos (5'-GCGACTCCGAGCGTGTGCA TGATG-3', 5'-ATGTCAAATCAAGGTAATAATCCAC-3'; Genetools) was used to knockdown translation of Ndr1 and Ndr2. For all knockdowns, a morpholino mixture of ~1.8 nl was injected into the yolk of one cell stage embryos. All morpholino mixtures contained Danieau's Buffer and 0.5 mg/ml phenol red.

### RNA rescue injections and overexpression

The pCS2(+)-*gdf3*-WT construct was generated by inserting the complementary sequence of WT *gdf3* mRNA into a pCS2(+) plasmid construct. The *gdf3* insert was amplified from cDNA using primers with a 5'-end BamHI site (5'-AAGGATCCTGTTTTTATAATCTAATAATGTTCTTGG-3') and a 5'-end XbaI site (5'-AATCTAGAATAGTAAAAGTTTTTATTATTACATTACAATG-3'). The amplicon was gel purified using a QIAquick Gel Extraction Kit (Qiagen, 28704) and both the *gdf3* insert and the pCS2(+) destination plasmid were digested with BamHI (NEB, R0136S) and XbaI (NEB, R0145S). The digested DNA was purified using a Qiaquick PCR Purification Kit (Qiagen, 28104) and ligated using T4 DNA ligase (NEB, M0202S). The resulting plasmid was sequenced to confirm the insertion of

*gdf3* into the multiple cloning site of pCS2(+).The plasmid was digested with *XbaI*, and capped *gdf3* mRNA was synthesized using an mMessage mMachine SP6 Transcription Kit. A 1.8 nl volume *gdf3* mRNA in 0.1 M KCl was injected into the cell of a one cell stage M*gdf3* or MZ*gdf3* embryo. RNA *in situ* hybridization was used to analyze *gsc, lft1, ntl,* and *sox17* expression. Nodal ovexpression was performed using *nodal* mRNA generated from a pCS2(+) plasmid, given to us as a gift from the Sampath Lab, containing the sequence for *ndr1* linked to the sequence for 3XFlag (*Wang et al., 2016*). Capped *nodal* mRNA was synthesized and injected similar to the *gdf3* mRNA rescues.

## RNA *in situ* hybridization

Embryos were fixed at the 10 ss stage in 4% PFA overnight at 4°C. These embryos were washed with in PBST (1X PBS containing 0.1% Tween-20), dechorionated, transitioned to 100% methanol, and stored at −20°C for at least 1 day. The transition to methanol was done by performing five-minute washes in 75% 1X PBST:25% methanol, 50% 1X PBST:50% methanol, 25% 1X PBST:75% methanol, and 100% methanol. The embryos were then transitioned into 1X PBST by performing 5 min washes in 25% 1X PBST:75% methanol, 50% 1X PBST:50% methanol, and 75% 1X PBST:25% methanol. Embryos were then washed four times in 1X PBST with 5 min per wash. Somite stage embryos were incubated for 1 min in 1X PBST containing 0.01 mg/ml Proteinase K (Sigma-Aldrich, P2308) followed by a 20 min incubation in 1X PBST containing 4% paraformaldehyde. These embryos were then washed five more times in 1X PBST with 5 min per wash. Blastula and gastrula stage embryos did not undergo this Proteinase K treatment, extra fixation with 4% paraformaldehyde, or the extra five washes with 1X PBST. Embryos were incubated in HYB (50% formamide, 5X SSC, 500 µg/ml torula yeast RNA, 50 µg/ml heparin 0.1% Tween 20, and 9 mM Citric Acid (pH 6.0)) for 2 hr at 68°C. Embryos were then incubated overnight in HYB containing an insitu hybridization probe at 68°C. The next day, embryos were washed at 68°C in HYB, 75% HYB: 25% 2X SSC, 50% HYB: 50% 2X SSC, 25% HYB: 75% 2X SSC, and 2X SSC for 10 min each wash. Embryos were then washed twice in 0.2X SSC for 30 min each wash. The remaining washes were performed at room temperature. Embryos were washed in 75% 0.2X SSC: 25% 1X PBST, 50% 0.2X SSC: 50% 1X PBST, 25% 0.2X SSC: 75% 1X PBST, and 1X PBST for 5 min each wash. Next, embryos were incubated on a rocker for 2 hr in 1X PBST containing 2 mg/ml BSA and 2% normal sheep serum (NSS). Embryos were then incubated overnight on a rocker in 1X PBST containing 2 mg/ml BSA, 2% NSS, and 1:3500 of Anti-Digoxigenin-AP (Roche, 11093274910). The next day, the embryos were washed quickly in 1X PBST followed by six additional 15 min 1X PBST washes on a rocker. Embryos were then washed three times in NTMT (0.1 M Tris-Cl ph 9.5, 0.1 M NaCl, 0.05 M MgCl$_2$, 0.1% Tween 20) and stained with 5 µl of NBT (Roche, 11383213001) and 3.75 µl BCIP (Roche, 11383221001) per 1 ml of NTMT. Staining was stopped by washing the embryos three times with NTMT, a 5 min wash with 1X PBST, and a 4°C overnight incubation in 1X PBST containing 4% paraformaldehyde. The embryos were then transitioned to methanol using the same four-step PBST:methanol washes listed above. Embryos were stored in −20°C or cleared in 2:1 Benzyl Benzoate:Benzyl Alcohol prior to imaging. Canada Balsam containing 10% methyl salicylate was used to mount cleared embryos on a slide. RNA *in situ* hybridization staining was visualized using a Leica DMRA2 microscope and images were acquired using a Leica DFC450 C camera. The following probes were used for the *in situ* hybridizations: *gdf3* (*Helde and Grunwald, 1993*), *nodal-related 2* (*Rebagliati et al., 1998*), *lefty1* (*Bisgrove et al., 1999*), *goosecoid* (*Stachel et al., 1993*), *floating head* (*Talbot et al., 1995*), *no tail* (*Schulte-Merker et al., 1992*), *axial* (*Strähle et al., 1993*), *sox17* (*Alexander and Stainier, 1999*), *southpaw* (*Long et al., 2003*), *dand5* (*Hashimoto et al., 2004*), *myl7* (*Yelon et al., 1999*).

## Immunofluorescence

### Immunofluorescence for Bmp signaling

Embryos were fixed at 75% epiboly stage in 4% PFA overnight at 4°C. These embryos were washed in PBST (1X PBS containing 0.1% Tween-20), transitioned to 100% methanol, and stored at −20°C for at least 1 day. The transition to methanol was done by performing five-minute washes in 75% PBST:25% methanol, 50% PBST:50% methanol, 25% PBST:75% methanol, and 100% methanol. The embryos were then transitioned into PBST by performing 5 min washes in 25% PBST:75% methanol, 50% PBST:50% methanol, 75% PBST:25% methanol, and 100% PBST. After three more five-minute washes in PBST, the embryos were washed once more in PBDT (PBST containing 1% DMSO) after

which they were incubated in blocking solution of PBDT containing 10% normal goat serum (NGS) for 2 hr. Then, 1:100 anti-phosphorylatedSMAD1/5/8 antibody (Cell Signaling Technologies, 9511S) was added and incubated overnight at 4°C. The embryos were washed in PBDT containing 1% NGS and 0.1M NaCl for 1 min followed by six more 30 min washes at room temperature, after which the embryos underwent an overnight incubation in PBDT containing 10% NGS and 1:400 Donkey anti-Rabbit Alexa Fluor 647 secondary antibody (Invitrogen, A31573) at 4°C. After DAPI was added and the sample was rocked at room temperature for 20 min, the embryos were then washed in PBDT containing 1% NGS and 0.1M NaCl for one minute followed by five 30 min washes. The final PBDT wash was for 30 min.

## Imaging
The embryos were positioned with the animal side toward the coverslip of a 30 mm dish (MatTek, P35G-1.5–14 C) in 1.5% low-melt agarose. All the WT and mutant embryos were imaged in one session on the Nikon A1 inverted confocal microscope using the same parameters.

## Immunofluorescence for KV morphology
Immunofluorescence to examine KV morphology was performed with small changes to the protocol listed above for Bmp signaling. The embryos were blocked in PBDT containing 10% NSS (blocking solution). The first antibody incubation was performed in blocking solution containing 1:200 anti-PKCzeta (aPKC) (Santa Cruz Biotechnology, SC-216) primary antibody. The second antibody incubation was performed in PBDT containing 10% NSS and 1:400 Goat anti-Rabbit Alexa Fluor 647 (Invitrogen, A21246) secondary antibody. The subsequent washes were performed in PBDT containing 1% NSS and 0.1M NaCl. Analysis of KV morphology was performed in a Tg(sox17: GFP)[s870] line (*Sakaguchi et al., 2006*).

## Imaging
The embryos were positioned with the posterior end toward the coverslip (KV facing down) of a 30 mm dish (MatTek, P35G-1.5–14 C) in 1.5% low-melt agarose. All the uninjected and morpholino injected embryos were imaged in one session on the Nikon A1 using the same parameters.

## Quantification
### Quantification of phosphorylated SMAD1/5/8
The resulting z-stacks were opened in Imaris and positioned so that the D-V axis of the embryo was parallel to the left-right axis of the screen using the Free Rotate command. Snapshots in grayscale were taken of the DAPI channel and the Cy5 channel individually. For each embryo, a rectangle 200 pixels high and 650 pixels wide was taken so as to completely cover the middle of the embryo and the Plot Profile command was used to extract intensity of the DAPI and Cy5 channels along the D-V axis. A MATLAB script was used to align the intensity profiles and report the Cy5/DAPI ratio along the embryo from the ventral side (0) to the dorsal side (1). See provided source files for MATLAB script and data sourse files: *Figure 3—source code 1*, *2*, *3*.

### Quantification of *eve1* domain extension
For the WT and mutant images, a combination of red, green, and blue pixel values was chosen to threshold the set of images in MATLAB, so as to detect the purple *in situ* signal in an unbiased manner. Given an input image, the MATLAB script also produced a modified image with the *in situ* signal in white and the rest of image darkened, which was used for the next step. In ImageJ, an oval was manually traced out around the embryo and the extent of the circumference of embryo with *in situ* signal was manually identified using the Oval_Profile plugin with Number of Points as 360 and Analysis mode as Radial Sums. From the plot, the extent of degrees that the *in situ* stain occupied was measured and divided by 360 to get extent of the expression domain. See supplement for MATLAB script (*Figure 3—source code 1*, *2* and *3*).

## Acknowledgements

We are grateful for the exemplary collegiality of Alex Schier and Joe Yost in coordinating publications. We thank Victoria L Patterson and Daniel T Grimes for the constructive comments, helpful advice, and assistance in editing the manuscript. We thank Philip Johnson for zebrafish care, and Dr. Gary Laevsky and the Molecular Biology Confocal Microscopy Facility, which is a Nikon Center of Excellence, for microscopy support. GAJ acknowledges support from National Science Foundation Graduate Research Fellowship Grant DGE 1148900. This work was supported by funding awarded to RDB from the National Institute of Child Health and Development, grant R01HD048584. The content of this manuscript is solely the responsibility of the authors and does not necessarily represent the official views of the National Institutes of Health.

## Additional information

### Funding

| Funder | Grant reference number | Author |
| --- | --- | --- |
| Eunice Kennedy Shriver National Institute of Child Health and Human Development | R01HD048584 | Rebecca D Burdine |
| National Science Foundation | Graduate Research Fellowship DGE 1148900 | Granton A Jindal |

The content of this manuscript is solely the responsibility of the authors and does not necessarily represent the official views of the National Institutes of Health or National Science Foundation.

### Author contributions

Jose L Pelliccia, Conceptualization, Data curation, Formal analysis, Investigation, Methodology, Writing—original draft, Writing—review and editing; Granton A Jindal, Formal analysis, Validation, Investigation, Visualization, Methodology, Writing—review and editing; Rebecca D Burdine, Conceptualization, Supervision, Funding acquisition, Writing—original draft, Project administration, Writing—review and editing

### Author ORCIDs

Jose L Pelliccia http://orcid.org/0000-0002-8452-2126
Granton A Jindal http://orcid.org/0000-0003-2803-9831
Rebecca D Burdine http://orcid.org/0000-0001-6620-5015

### Ethics

Animal experimentation: Zebrafish husbandry was performed in strict accordance with the recommendations in the Guide for the Care and Use of Laboratory Animals. All protocols are approved (Burdine #1915-16) by Princeton's Institutional Animal Care and Use Committee (IACUC).

### Decision letter and Author response

Decision letter https://doi.org/10.7554/eLife.28635.024
Author response https://doi.org/10.7554/eLife.28635.025

## Additional files

### Supplementary files

• Transparent reporting form
DOI: https://doi.org/10.7554/eLife.28635.022

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
