## [Decision Letter]

Thank you for submitting your article "Gdf3 is required for robust Nodal signaling during germ layer formation and left-right patterning" for consideration by *eLife*. Your article has been reviewed by three peer reviewers, and the evaluation has been overseen by a Reviewing Editor and Didier Stainier as the Senior Editor. The reviewers have opted to remain anonymous.

The reviewers have discussed the reviews with one another and the Reviewing Editor has drafted this decision to help you prepare a revised submission.

Summary:

In this manuscript, Pelliccia and coauthors uncover an essential function of Gdf3/Vg1 in Nodal signaling by generating maternal zygotic mutants in zebrafish. They show that MZ*gdf3* mutants phenocopy previously described mutants for Nodal ligands (*cyc, sqt*) and coreceptor (MZ*oep*) and that partial knock-down of *gdf3* translation via morpholinos (MO) impairs Kupffer's Vesicle (KV) development and left-right (LR) asymmetry. It is notable that Vg1, the mouse homologs Gdf1/3, and zebrafish Gdf3 have been studied before, and that it has already been shown that these factors can bind to Nodal and enhance Nodal function. It has also been shown that Gdf1,3/Vg1 is required for early development, specifically for mesoderm development. The current work expands on these findings, giving more detail on some aspects, and, importantly, uses mutants rather than antisense approaches (as previously used in *Xenopus* and in zebrafish) to reach its conclusions, giving the current work a stronger basis than the prior work. Since Nodal signaling is fundamental in cell fate specification and organogenesis, these findings are of great interest to a broad audience. This work is carefully executed and clearly presented, and only a few points need adjustment during revision of this manuscript, as described below.

Essential revisions:

1) The authors find residual Nodal activity in MZ*gdf3* mutants and therefore conclude that *gdf3* facilitates Nodal signaling. However the fact that Nodal signaling is at all possible without *gdf3* could be shown more clearly. The authors could, for example, over-express nodal ligands in the MZ*gdf3* mutants and show to what extent signaling can be recovered compared to over-expression in the WT.

2) The ubiquitous expression of *gdf3* at early stages of development suggests that *gdf3* may act as a facilitator of Nodal signaling, while the patterning of mesendoderm tissues is achieved by localized expression of *cyc* and *sqt*. This point could be better shown by more precise rescue experiments. Is it possible to obtain fully patterned embryos by simple over-expression of *gdf3* in MZ*gdf3* mutants? The experiment in Figure 2 suggests that *gsc* expression can be recovered but that gsc localization remains aberrant.

3) The experiment on KV morphology and LR asymmetry are interesting as they show that *gdf3* acts as a facilitator of all known Nodal ligands in zebrafish. However the authors should confirm their findings with a rescue of the MO phenotype they show or, alternatively, by obtaining similar results by partial rescue of MZ*gdf3* mutant with *gdf3* mRNA.

4) The authors state in the paper several times that perhaps Nodals and Vg1 dimerize. Could this be shown directly (e.g., dimerization of Gdf3 and *southpaw*)? If so, this would boost the overall impact of the manuscript. However, similar findings have been previously shown for, for example, BMPs.

---

## [Author Response]

Essential revisions:1) The authors find residual Nodal activity in MZgdf3 mutants and therefore conclude that gdf3 facilitates Nodal signaling. However the fact that Nodal signaling is at all possible without gdf3 could be shown more clearly. The authors could, for example, over-express nodal ligands in the MZgdf3 mutants and show to what extent signaling can be recovered compared to over-expression in the WT.

We have performed the experiment the reviewer suggested. By overexpressing *ndr1 (squint*) via mRNA injections at the one-cell stage, we determined that higher amounts of Nodal could activate the Nodal target gene *goosecoid (gsc*) in MZ*gdf3* mutants. Although 1 pg of *ndr1* led to a modest increase in *gsc* expression in WT embryos it did not activate *gsc* expression in MZ*gdf3* embryos. However, 10pg of injected *ndr1* mRNA did lead to activation of *gsc* expression in MZ*gdf3* mutant embryos. This new data has been included in Figure 3 and suggests that Ndr1 can signal to activate genes independently of Gdf3 when it is present at very high levels.

2) The ubiquitous expression of gdf3 at early stages of development suggests that gdf3 may act as a facilitator of Nodal signaling, while the patterning of mesendoderm tissues is achieved by localized expression of cyc and sqt. This point could be better shown by more precise rescue experiments. Is it possible to obtain fully patterned embryos by simple over-expression of gdf3 in MZgdf3 mutants? The experiment in Figure 2 suggests that gsc expression can be recovered but that gsc localization remains aberrant.

We have addressed this comment by performing various rescue experiments in M/MZ*gdf3* embryos. mRNA generated from the previously published *gdf3* plasmid construct we received as a gift (Peterson et al.2013) did not rescue mutant embryos. This was likely due to the presence of an HA tag at the C-terminus of the protein. Therefore, we generated a new untagged *gdf3* plasmid construct from cDNA and performed rescue experiments using mRNA from this. We were able to obtain fully patterned embryos by overexpressing this untagged Gdf3 in MZ*gdf3* embryos and included this new data in Figure 1. This mRNA was also able to rescue Nodal target gene expression, data now included in Figure 2 and in Figure 2—figure supplement 1. In each case, there was correct spatial expression of each target gene. This suggests that Gdf3 is a permissive factor and is not required in any particular spatial domain for proper Nodal target gene induction.

3) The experiment on KV morphology and LR asymmetry are interesting as they show that gdf3 acts as a facilitator of all known Nodal ligands in zebrafish. However the authors should confirm their findings with a rescue of the MO phenotype they show or, alternatively, by obtaining similar results by partial rescue of MZgdf3 mutant with gdf3 mRNA.

We have addressed this comment by performing gdf3 mRNA overexpression experiments in M*gdf3* and MZ*gdf3* embryos in which early mesendoderm defects are rescued. Defects in left-right patterning (a later event) were still present to varying degrees. Increasing amounts of injected gdf3 mRNA led to increasing rescue of L-R defects in MZ*gdf3* mutants, as assessed by the direction of cardiac jogging and the laterality of *spaw* and *dand5* expression. Moreover, M*gdf3* mutants, in which zygotically-derived Gdf3 is unperturbed, could be rescued for L-R patterning to a greater extent than MZ*gdf3* mutants. These new results are presented in Figure 5, and confirm our initial conclusion that Gdf3 acts as a facilitator of Nodal signaling during L-R patterning events. Moreover, the difference in "rescuability" of MZ*gdf3* and M*gdf3* mutants suggests some role for zygotic Gdf3 in L-R patterning that is not revealed in Z*gdf3* mutants likely owing to compensation by maternal Gdf3. This is a curious result, and we added some thoughts on this in our Discussion.

4) The authors state in the paper several times that perhaps Nodals and Vg1 dimerize. Could this be shown directly (e.g., dimerization of Gdf3 and southpaw)? If so, this would boost the overall impact of the manuscript. However, similar findings have been previously shown for, for example, BMPs.

We appreciate this suggestion and agree that testing for direct interaction between Gdf3 and zebrafish Nodal orthologs would be informative. Unfortunately, we are unable to supply these data in sufficient quality given the time constraints, and we hope the reviewers agree that this does not detract significantly from our manuscript. We also note that Nodal-Gdf3 heterodimerization has been demonstrated in other systems (Fuerer et al., 2014; Tanaka et al.2007), something we have emphasized in our manuscript.